# AdaSPEC: Selective Knowledge Distillation for Efficient Speculative Decoders

**Yuezhou Hu**[1*], **Jiaxin Guo**[2*], **Xinyu Feng**[3], **Tuo Zhao**[3†]

[1] University of California, Berkeley  [2] Tsinghua University
[3] Georgia Institute of Technology

yuezhouhu@berkeley.edu  jx-guo21@mails.tsinghua.edu.cn
{xfeng300,tourzhao}@gatech.edu

## Abstract

Speculative Decoding (SD) accelerates large language model inference by employing a small draft model to generate predictions, which are then verified by a larger target model. The effectiveness of SD hinges on the alignment between these models, which is typically enhanced by Knowledge Distillation (KD). However, conventional KD methods aim to minimize the KL divergence between the draft and target models across all tokens, a goal that is misaligned with the true objective of SD, which is to maximize token acceptance rate. Therefore, draft models often struggle to fully assimilate the target model's knowledge due to capacity constraints, leading to suboptimal performance. To address this challenge, we propose AdaSPEC, a novel method that incorporates selective token filtering into the KD process. AdaSPEC utilizes a reference model to identify and filter out difficult-to-fit tokens, enabling the distillation of a draft model that better aligns with the target model on simpler tokens. This approach improves the overall token acceptance rate without compromising generation quality. We evaluate AdaSPEC across diverse tasks, including arithmetic reasoning, instruction-following, coding, and summarization, using model configurations of 31M/1.4B and 350M/2.7B parameters. Our results demonstrate that AdaSPEC consistently outperforms the state-of-the-art DistillSpec method, achieving higher acceptance rates across all tasks (up to 15%). The code is publicly available at https://github.com/yuezhouhu/adaspec.

## 1 Introduction

Large language models (LLMs) have revolutionized natural language processing, achieving impressive performance across a wide range of tasks. Models like GPT-4 [25] and Llama 3 [11] demonstrate state-of-the-art results in various natural language understanding and generation tasks [2, 27], including highly complex tasks such as summarization [23] and mathematical reasoning [9, 13]. However, as these models grow in size and complexity, their inference becomes increasingly computationally intensive, leading to practical challenges in deployment, including slow generation speeds and significant output latency.

To address these shortcomings, current approaches primarily focus on achieving a trade-off between efficiency and performance through two main strategies. The first involves compressing the model scale to enhance the capability of smaller models, often using techniques like Knowledge Distillation (KD) [14]. The second approach employs methods such as quantization to enable faster computation. However, these strategies inevitably lead to a sacrifice of performance, either due to a loss of representational capacity during compression or reduced accuracy resulting from optimization for

---

[*]Equal Contribution. Work done during an internship at Georgia Tech.  [†]Corresponding author.

39th Conference on Neural Information Processing Systems (NeurIPS 2025).

speed. As a result, there is a growing need for methods that can maintain the high performance of LLMs while significantly improving their inference efficiency.

Recently, Speculative Decoding (SD) [16, 7] has emerged as a promising paradigm for accelerating LLM inference without sacrificing performance. Unlike model compression or quantization, which modify the model architecture or parameters, SD accelerates generation by restructuring the decoding process itself. Specifically, it introduces a lightweight draft model that speculatively generates multiple candidate tokens, which are then verified by the larger target model. This paradigm preserves the target model's predictive quality while substantially reducing the number of expensive forward passes, offering a new efficiency–performance trade-off.

The core of SD lies in the design of the draft model. This model is typically much smaller than the target model, even ranging from one-tenth to one-hundredth of the size, enabling faster token generation while maintaining a certain level of capability. Consequently, the actual inference speed-up achieved by SD relies on the draft model closely aligning its predictions with the target model's output distribution. Typically, this alignment is achieved by pre-training and fine-tuning both models on the same datasets, yielding a pair of homogeneous models from the same family, sharing the same architecture but differing in size. However, training two models on the same datasets does not necessarily produce optimal alignment, especially given the significant scale disparity between the draft and target models. This difference in scale makes the draft model prone to prediction errors. To address this challenge, state-of-the-art methods employ KD techniques to refine the draft model, rather than relying solely on direct fine-tuning [32]. However, optimizing fidelity metric (e.g., forward KL divergence) does not necessarily lead to a high acceptance rate. Worse still, it may waste the draft model's limited capacity on tokens that are inherently hard to learn and unlikely to be accepted anyway. Additionally, these methods may encounter issues such as the loss failing to converge. Given these challenges, there is a critical need for SD-specific training regimes that effectively balance model capacity constraints with prediction accuracy requirements.

Fortunately, we observe substantial variation in the difficulty of learning individual tokens during KD, which has critical implications for transferring knowledge from the teacher (target) model to the student (draft) model. Instead of mimicking the full output distribution of the target model, the draft model only needs to produce correct predictions on the subset of tokens that is easy enough to propose. During the process of distillation, we identify a subset of "hard" tokens that pose particular challenges for the student model to learn and to predict accurately, regardless of training efforts. Conversely, other tokens are relatively easy to assimilate. We argue that uniformly emphasizing the loss on both "easy" and "hard" tokens may be counterproductive. Attempting to reduce the loss on difficult tokens often comes at the expense of increasing the loss on easy tokens, resulting in suboptimal learning across both categories. To address this issue, we propose a novel approach: deliberately excluding "hard" tokens from the training process. By focusing the loss function exclusively on "easy" tokens, we can more effectively utilize the limited capacity of the student model, thus achieving better alignment with the teacher model on these tokens. This strategic exclusion of hard-to-learn tokens allows the student model to concentrate its resources on mastering the more accessible aspect of the teacher's knowledge, potentially leading to improved overall performance in SD tasks. Our approach thus maximizes the alignment between the draft and target models within the constraints of the draft model's capacity.

In this study, we propose **AdaSPEC**, a novel Knowledge Distillation method designed to bridge the capacity gap between the draft and target models in SD. AdaSPEC operates in two phases:

**1:** Reference Model Distillation and Token Filtering: A reference model, initialized as a copy of the draft model, is distilled using the target model as its teacher. For simplicity, we assume that the target model has been well fine-tuned to downstream tasks of our interests. Here the reference model serves a crucial role as a token filter. It identifies "hard" tokens—those that are difficult for smaller models to predict accurately—by comparing the perplexity differences between the reference and draft models on the training data.

**2:** Selective Draft Model Distillation: Finally, the draft model undergoes distillation using a filtered dataset. The reference model removes the previously identified "hard" tokens, allowing the draft model to focus its limited capacity on learning to predict the remaining, more manageable tokens accurately.

We conduct extensive experiments on a wide range of models and downstream tasks, where we benchmark AdaSPEC against DistillSpec and find that AdaSPEC sucessfully pushes the limit of SD—across all tasks and model setups, AdaSPEC consistently achieves higher acceptance rates (up to 15%; see Table 1).

## 2  Preliminaries

In this section, we provide a formal overview of the foundational concepts. We begin with the mathematical framework of SD, followed by a description of various evaluation metrics. Finally, we explore language model families and their significance in enabling techniques like SD to bridge performance gaps between models of different sizes.

**Speculative Decoding.** Speculative Decoding [7, 16, 29, 20, 32, 6, 17, 18, 30, 6, 21, 26, 32, 20] is originally proposed to accelerate LLM inference by employing a compact draft model to predict potential output sequences in advance and then verified by a larger target model. The typical framework of SD is formulated as follows. Let $M_p$ and $M_q$ denote the large target model and the compact draft model, respectively. SD leverages the draft model to autoregressively generate $\gamma$ tokens $\boldsymbol{z} \triangleq \{z_i\}_{i=1}^{\gamma} \sim q_\theta(\cdot \mid \boldsymbol{x})$ based on the input $\boldsymbol{x} = [x_1, x_2, \ldots, x_t]$, which includes the prompt and previously generated tokens. The target model then verifies these proposed tokens by evaluating their probabilities $\{p(z_i \mid \boldsymbol{x}, \boldsymbol{z}_{<i})\}_{i=1}^{\gamma}$ in parallel.

Both models generate probability distributions $p(z_{i+1} \mid \boldsymbol{x}, \boldsymbol{z}_{<i})$ and $q(z_{i+1} \mid \boldsymbol{x}, \boldsymbol{z}_{<i})$ for each token $i = 1, \ldots, \gamma$ in a single forward pass. Using a greedy decoding strategy, only the tokens with the highest probabilities are selected for generation or verification. The sampling functions are:

$$S_p(\boldsymbol{z}_{<i}) = \arg\max_{z_{i+1}} p(z_{i+1} \mid \boldsymbol{x}, \boldsymbol{z}_{<i}), \tag{1}$$

$$S_q(\boldsymbol{z}_{<i}) = \arg\max_{z_{i+1}} q(z_{i+1} \mid \boldsymbol{x}, \boldsymbol{z}_{<i}), \tag{2}$$

for each $i = 1, \ldots, \gamma$. The complete sampling and verification process is detailed in Appendix A.1

**Acceptance Rate.** The acceptance rate, $\alpha$, measures the accuracy of the draft model $M_q$ compared to the target model $M_p$. It is calculated as:

$$\alpha = \frac{\text{accept}}{\text{accept} + \text{reject}}. \tag{3}$$

Here, $\text{accept}$ and $\text{reject}$ are the count of tokens accepted and rejected by $M_p$, respectively. A higher $\alpha$ indicates greater alignment between $M_p$ and $M_q$, facilitating faster inference in practical scenarios.

**Block Efficiency and Wall-time Improvement.** Block efficiency [7, 16], $\tau$, quantifies the average number of tokens generated per iteration. It is defined as the expected number of accepted tokens per block, with a maximum value of $\gamma + 1$ for a block size of $\gamma$. The block efficiency can also be expressed in terms of the acceptance rate $\alpha$ [16]:

$$\tau(x) = \frac{1 - \alpha^{\gamma+1}}{1 - \alpha}. \tag{4}$$

This metric evaluates how effectively $M_q$ approximates $M_p$. The speed-up factor for the total wall-time is given by:

$$\text{Speed-up} = \frac{\tau(x)}{\gamma c + 1}, \tag{5}$$

where $c$ is the cost coefficient, representing the ratio of the time taken by a single execution of $M_q$ to that of $M_p$.

**Language Model Families.** Modern language models are often developed as part of a family of models that share the same core architecture but differ in scale, typically measured by the number of parameters or the size of the training dataset. These families, such as Llama 3 [11], BERT [10], and Pythia [5], are designed to enable researchers and practitioners to balance computational efficiency and performance based on specific use cases. Within a family, smaller models are generally used for tasks requiring faster inference or lower computational cost, while larger models are leveraged for tasks demanding higher accuracy and richer representations. This structural consistency within a family allows for techniques like Knowledge Distillation [14] and Speculative Decoding [7, 16] to transfer knowledge or align predictions effectively between models of varying sizes.

## 3 Method

We introduce AdaSPEC, an adaptive distillation framework for SD that enhances the alignment between a target model and a smaller draft model through selective Knowledge Distillation. Given a target model $M_p$ fine-tuned for a specific downstream task, AdaSPEC consists of two key steps: (1) constructing a reference model $M_{\text{ref}}$ and (2) selectively distilling knowledge from $M_p$ and $M_{\text{ref}}$ to the draft model $M_q$.

**Step 1: Constructing the Reference Model.** The reference model $M_{\text{ref}}$ is constructed by distilling $M_p$ on a downstream task dataset $\mathcal{D}$ using the DistillSpec framework [32]. The objective is to minimize the forward KL divergence between the target model and the reference model:

$$\mathcal{L}_{\text{KD}} = \mathbb{E}_{\boldsymbol{x} \sim \mathcal{D}, \boldsymbol{y} \sim P(\boldsymbol{y}|\boldsymbol{x})} \left[ \mathcal{K} \left( P(\boldsymbol{y}|\boldsymbol{x}) || R(\boldsymbol{y}|\boldsymbol{x}) \right) \right], \tag{6}$$

where $\boldsymbol{x}$ represents the input prefix, $\boldsymbol{y}$ denotes the generated context, $\mathcal{K}$ denotes the forward KL divergence, $P(\boldsymbol{y}|\boldsymbol{x})$ represents the probability distribution of the target model, and $R(\boldsymbol{y}|\boldsymbol{x})$ corresponds to the probability distribution of the reference model.

**Step 2: Selective Knowledge Distillation for the Draft Model.** To identify learnable tokens for the draft model $M_q$, we compute token-wise losses based on the predicted distributions of $M_{\text{ref}}$ and $M_q$. Specifically, for each token $w$, the token-wise KL divergence losses are computed as:

$$\mathcal{L}_{\text{ref}}(w) = \mathcal{K} \left( P(w \mid \text{context}) || R(w \mid \text{context}) \right), \tag{7}$$

$$\mathcal{L}_{\text{draft}}(w) = \mathcal{K} \left( P(w \mid \text{context}) || Q(w \mid \text{context}) \right), \tag{8}$$

where $Q(w \mid \text{context})$ is the probability predicted for token $w$ by the draft model, given the context.

Next, we calculate the difference in token-wise losses:

$$\Delta_{\mathcal{L}}(w) = \mathcal{L}_{\text{draft}}(w) - \mathcal{L}_{\text{ref}}(w). \tag{9}$$

Tokens with higher $\Delta_{\mathcal{L}}(w)$ represent a larger performance gap between $M_q$ and $M_{\text{ref}}$ relative to $M_p$, suggesting that these tokens are **not yet well aligned** but are **highly learnable** for the draft model. Accordingly, we select the subset of tokens with larger $\Delta_{\mathcal{L}}(w)$ values, as they are most promising for improving the alignment between the draft and target models. Specifically, we denote

$$\mathbb{S} = \{ w \mid \Delta_{\mathcal{L}}(w) \text{ is among the top } k \times 100\% \text{ of all tokens} \}, \quad k \in [0, 1].$$

Therefore, the overall loss for training the draft model $M_q$ is:

$$\mathcal{L}_{\text{distill}} = \frac{1}{k \cdot |\boldsymbol{y}|} \sum_{i=1}^{|\boldsymbol{y}|} \mathbb{I} \left[ y_i \in \mathbb{S} \right] \cdot \mathcal{L}_{\text{draft}}(y_i), \tag{10}$$

where $\mathbb{I}[\cdot]$ is the indicator function that equals 1 if the condition inside the brackets is satisfied, and 0 otherwise. It ensures that only the selected learnable tokens contribute to the loss calculation. The whole filtering process is shown in figure 1.

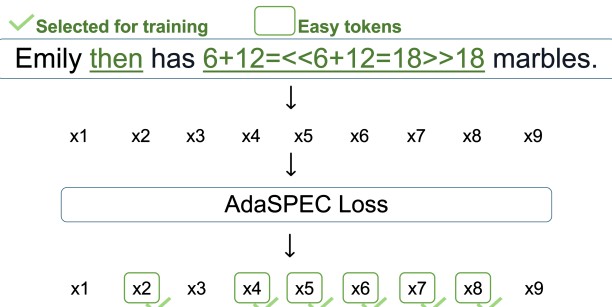

Figure 1: **Overview of AdaSPEC distillation process**: AdaSPEC selects the most training-effective tokens and distills on these tokens.

# 4 Experiments

We evaluate AdaSPEC through comprehensive experiments across diverse domains and conduct detailed ablation studies to analyze its impact on the acceptance rate $\alpha$.

## 4.1 Experimental Setup

Our experimental framework employs GPT-like decoder-only Transformer models in two distinct configurations, designed to evaluate performance across different parameter scales while maintaining tokenizer consistency for SD:

• **Small-to-Large Model Configuration:** A draft model *Pythia-31M* paired with target model *Pythia-1.4B* [5]. These models share architecture and tokenizer, providing an ideal test case for same-family knowledge transfer.

• **Medium-to-Large Model Configuration:** A draft model *CodeGen-350M* paired with target model *Phi-2* [24, 1]. While from different families, these models use an aligned tokenizer to ensure token-level consistency, allowing us to evaluate cross-family KD.

We test these two configurations on a diverse set of five tasks, each representative of a specific domain to provide a robust evaluation framework for AdaSPEC: GSM8K [9] (A benchmark for multi-step arithmetic reasoning), Alpaca [27] (A comprehensive instruction following dataset), MBPP [3] (A Python programming challenge set for code generation), CNN/Daily Mail [22] (A long-form summarization task), and XSUM [23] (An extreme summarization challenge).

**Reference Model Training.** To ensure a consistent starting point and fair comparison, both the draft model and the reference model are initialized from the same pre-trained model. For each task, we first fine-tune the target model on the task-specific dataset to establish a strong baseline. The reference model is then trained using the method from DistillSpec [32].

## 4.2 Baseline Setup

We compare AdaSPEC against DistillSpec [32], the current state-of-the-art method for SD. Although AdaSPEC builds upon DistillSpec's training framework for its reference model, it introduces novel token selection mechanism. To evaluate its effectiveness, we evaluate both methods under two settings: a resource-efficient scenario with fixed training duration and a scenario optimized for maximum performance:

• **3-Epoch Setting:** Both reference and draft models are trained for exactly 3 epochs, a standard practice in LLM fine-tuning that balances task-specific performance with general capability retention [4]. This controlled training duration effectively prevents overfitting while ensuring adequate task adaptation. This setting evaluates model effectiveness under typical resource constraints and provides insights into rapid adaptation scenarios.

• **Optimal-Epoch Setting:** Models are trained for a variable number of epochs, treated as a tunable hyperparameter, to maximize task-specific performance. While this approach may lead to overfitting to the specific task at the expense of performance on other tasks, it allows us to thoroughly evaluate the upper bound of performance. The optimal number of epochs is determined empirically. Specifically, for GSM8K, the number of target epochs is chosen according to validation accuracy, while for the rest of the experiments it is chosen according to validation perplexity. Afterwards, we distill the reference model and pick the one with highest $\alpha$ on validation set. Eventually, this model serves as reference to train our draft model. For robustness, we only select the optimal epoch from 1, 3, 6, 10, 15, 20 and 30 (for XSUM and CNN/Daily Mail we select from 1, 3, 6, 10 for training efficiency). This configuration enables evaluation of both methods under less constrained scenarios, where achieving optimal task performance takes precedence over maintaining general capabilities.

While Zhou et al. [32] employs a more extensive training schedule in their DistillSpec experiments, our study adopts a more resource-efficient approach due to computational constraints. In the **Optimal-Epoch Setting**, we limit training to a maximum of 30 epochs, striking a balance between performance optimization and computational feasibility. Complete hyperparameter configurations and training specifications for both DistillSpec and AdaSPEC are detailed in Appendix A.2.

## 4.3 Main Results

We summarize the main experimental results in Table 1.

Table 1: Main experimental results for AdaSPEC compared to DistillSpec under two settings: 3-Epoch and Optimal-Epoch. Metrics include acceptance rate ($\alpha$).

| Task | 3-Epoch ($\alpha$) | | | | Optimal-Epoch ($\alpha$) | | | |
|---|---|---|---|---|---|---|---|---|
| | Pythia-31M $\to$ 1.4B | | CodeGen-350M $\to$ Phi-2 | | Pythia-31M $\to$ 1.4B | | CodeGen-350M $\to$ Phi-2 | |
| | DistillSpec | AdaSPEC | DistillSpec | AdaSPEC | DistillSpec | AdaSPEC | DistillSpec | AdaSPEC |
| GSM8K | 57.58% | **62.63%** | 79.49% | **82.79%** | 66.19% | **68.28%** | 81.49% | **83.48%** |
| Alpaca | 44.34% | **47.25%** | 56.48% | **58.80%** | 65.41% | **65.79%** | 58.05% | **60.36%** |
| MBPP | 46.88% | **47.73%** | 87.36% | **88.76%** | 49.88% | **65.12%** | 86.60% | **87.70%** |
| CNN/Daily Mail | 73.05% | **74.22%** | 79.33% | **80.63%** | 80.15% | **80.89%** | 85.01% | **86.29%** |
| XSUM | 47.24% | **49.11%** | 58.88% | **59.93%** | 56.11% | **57.80%** | 66.78% | **68.19%** |

**Acceptance Rate Analysis.** We evaluate performance using the acceptance rate $\alpha$, defined as the proportion of draft-model-generated tokens validated by the target model. As shown in Table 1, AdaSPEC consistently achieves higher acceptance rates than DistillSpec across all tasks and model configurations, demonstrating superior draft-target model alignment.

## 4.4 Analysis

To provide detailed insights into AdaSPEC's effectiveness, we conduct in-depth analyses on two representative configurations:

• **Pythia-31M/1.4B on GSM8K (3-Epoch):** This configuration examines performance on arithmetic reasoning under constrained training conditions, representing scenarios with limited computational resources and the need for generalization. Since reasoning is typically considered as an additional capability beyond general language modeling, this setup ensures that the model retains its core abilities while effectively handling arithmetic tasks.

• **Pythia-31M/1.4B on CNN/Daily Mail (Optimal-Epoch):** This setup investigates extractive summarization with extended training, demonstrating the model's ability to optimize for task-specific objectives. In real-world applications, models are sometimes specifically deployed for summarizing long-form contents such as news reports, emails, or web pages, requiring dedicated fine-tuning. Thus, the Optimal-Epoch setting is chosen to maximize the model's summarization capabilities.

**Task-Level Acceptance Rate Distribution.** We first analyze the distribution of acceptance rates across tasks for both methods. As illustrated in Figure 2, AdaSPEC demonstrates consistently superior performance compared to DistillSpec. The acceptance rate histograms for both tasks exhibit a significant rightward shift under AdaSPEC, indicating more frequent successful draft predictions. This systematic improvement in acceptance rate suggests that AdaSPEC's selective distillation approach effectively enhances draft-target model alignment across diverse task contexts.

**Logit Margin Distributions Across Tokens.** Next, we analyze the distribution of top-2 logit margins across tokens for both methods. The logit margin, defined as the difference between the logits of the top-1 and top-2 predicted tokens, serves as a measure of prediction confidence. A positive margin indicates a correct draft model prediction, while a negative margin signifies an incorrect prediction that would be rejected in SD.

As shown in Figure 2, AdaSPEC demonstrates superior logit margin distributions compared to DistillSpec across both GSM8K and CNN/Daily Mail datasets. AdaSPEC exhibits:

• Higher frequency and magnitude of positive margins, indicating more frequent and confident correct predictions.

• Lower frequency and magnitude of negative margins, suggesting less frequent and less severe prediction errors.

These patterns demonstrate that AdaSPEC achieves better draft-target model alignment through its selective distillation approach, enabling more effective knowledge transfer from the target model to the draft model.

**KL-Divergence Distribution Across Tokens.** We further analyze the Kullback-Leibler (KL) divergence between draft and target models' token prediction distributions on both GSM8K and CNN/Daily Mail datasets. As illustrated in Figure 2, AdaSPEC exhibits consistently lower KL divergence values compared to DistillSpec across both tasks, demonstrated by significant leftward shifts in the distributions. This systematic reduction in KL divergence across different tasks and tokens indicates that AdaSPEC's selective distillation approach achieves tighter alignment between draft and target model predictions, corroborating our previous findings on acceptance rates and logit margins.

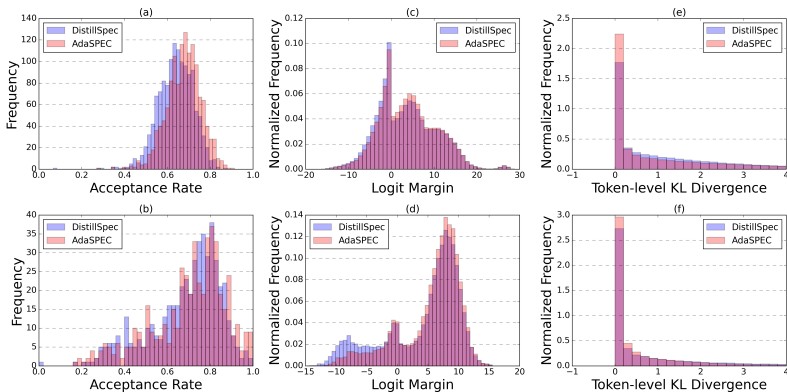

Figure 2: **Comparative analysis of AdaSPEC and DistillSpec performance across multiple metrics on GSM8K (a, c, e) and CNN/Daily Mail (b, d, f) datasets:** (a-b) Task-level acceptance rate distributions showing AdaSPEC's superior performance across tasks. (c-d) Logit margin distributions demonstrating AdaSPEC's improved prediction confidence with higher positive margins and lower negative margins. (e-f) Token-level KL divergence distributions indicating better draft-target model alignment for AdaSPEC with consistently lower divergence values. The results demonstrate AdaSPEC's more effective knowledge transfer and improved draft-target model alignment compared to DistillSpec across different evaluation metrics.

**Case Studies.** We conduct detailed case studies on GSM8K and CNN/Daily Mail datasets. A consistent pattern emerges: AdaSPEC's prediction errors form nearly a subset of DistillSpec's errors, as illustrated in Figure 3. This pattern demonstrates the general effectiveness of AdaSPEC's targeted training approach in improving alignment and reducing inference discrepancies.

GSM8K, with its natural division between mathematical and non-mathematical tokens, offers particularly insightful analysis. During training, AdaSPEC predominantly selects mathematics-related tokens for focused learning (see Appendix A.3). During inference, this selective approach translates into significantly improved prediction accuracy for mathematical tokens compared to DistillSpec, as shown in Figure 3. These results demonstrate AdaSPEC's ability to identify and prioritize task-critical tokens during training, leading to more precise draft-target model alignment.

### 4.5 Ablation Study

To systematically evaluate the effectiveness of different components in AdaSPEC, we conduct comprehensive ablation studies across the following four key dimensions. All experiments are conducted on GSM8K and MBPP with Pythia 1.4B (target) and Pythia 31M parameters (draft). All models are trained for 3 epochs.

**Token Selection Mechanism.** To evaluate our token selection strategy, we compare models trained on the top 40% of tokens (selected based on KL-divergence margin) against those trained on the bottom 40%. As shown in Table 2, models trained on the top 40% tokens consistently outperform those trained on the bottom 40%, with the latter performing even worse than the reference model. The improvement is particularly pronounced on the MBPP dataset, where token selection yields up to a 6% performance gain. These results demonstrate that AdaSPEC effectively enhances model alignment by focusing on more learnable tokens during Knowledge Distillation.

**Training Method.** To demonstrate that AdaSPEC's benefits extend beyond Knowledge Distillation, we replace the distillation process for both reference and draft models with direct fine-tuning. Table 3

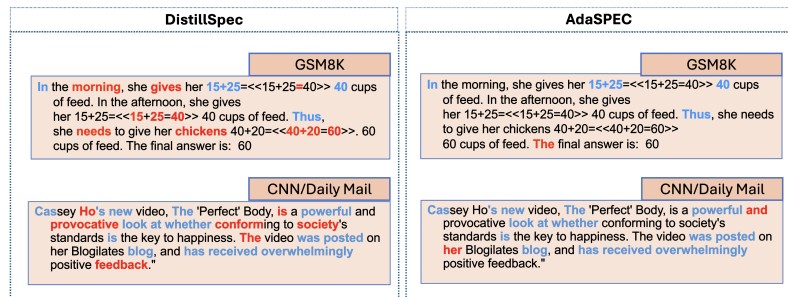

Figure 3: **Comparison of prediction errors between AdaSPEC and DistillSpec on GSM8K and CNN/Daily Mail Datasets:** Tokens highlighted in blue represent errors made by both methods, while tokens highlighted in red indicate errors unique to the corresponding method. As can be seen, AdaSPEC's errors form nearly a subset of DistillSpec's errors, demonstrating the effectiveness of AdaSPEC's selective training approach in reducing inference discrepancies.

Table 2: Ablation study results for token selection strategies.

| Sub-Strategy | GSM8K | | MBPP | |
| --- | --- | --- | --- | --- |
| | Reference $\alpha$ | Draft $\alpha$ | Reference $\alpha$ | Draft $\alpha$ |
| Top 40% | 59.77% | **63.22%** | 42.22% | **48.22%** |
| Bottom 40% | **59.77%** | 49.03% | **42.22%** | 39.75% |

Table 3: Ablation study results for training methods.

| Sub-Strategy | GSM8K | | MBPP | |
| --- | --- | --- | --- | --- |
| | Reference $\alpha$ | Draft $\alpha$ | Reference $\alpha$ | Draft $\alpha$ |
| Distillation | 59.77% | **63.22%** | 42.22% | **48.22%** |
| Fine-tuning | 59.64% | **63.13%** | 41.42% | **45.61%** |

Table 4: Ablation study results for distillation methods.

| Sub-Strategy | GSM8K | | MBPP | |
| --- | --- | --- | --- | --- |
| | Reference $\alpha$ | Draft $\alpha$ | Reference $\alpha$ | Draft $\alpha$ |
| KL | 59.77% | **63.22%** | 42.22% | **48.22%** |
| TVD | 9.32% | 9.09% | 3.86% | 6.76% |
| RKL | 30.22% | 30.05% | 13.17% | 15.61% |

reveals two key findings: (1) fine-tuned draft models outperform the distillation baseline (reference model) across both datasets, confirming that our token selection process aids model convergence; and (2) fine-tuned draft models achieve up to 4% improvement over their reference counterparts, indicating that our token selection mechanism's benefits generalize beyond distillation to broader training scenarios.

**Distillation Method.** We expand AdaSPEC to more distillation approaches: Reverse KL (RKL) and Total Variation Distance (TVD) [28]. With $k = 0.4$ for all methods, we observe that token selection significantly improves the acceptance rate by 6% on MBPP when using forward KL. However, when using RKL and TVD, the acceptance rate performance degrades. This is primarily attributed to the inherent limitations of RKL and TVD as distillation objectives, which struggle to effectively align the draft and target models in the context of SD. It is worth noting that DistillSpec [32] uses TVD as the distillation function with a batch size of 32 and a training step of 300,000. This prolonged training process not only requires substantial computational resources but also results in the problem of overfitting. Considering these factors, we ultimately select forward KL divergence as our distillation objective.

**Token Selection Ratio.** To investigate the impact of token selection ratio, we vary $k$ and compare the final acceptance rate of the draft model. Results in Fig 4 show that typically, lower $k$ values result in better final acceptance rate. To strike a balance between training efficiency and performance, we finally choose $k = 0.4$ in most cases.

## 4.6 Additional Experimental Results

**Wall Clock Speed-up.** To investigate AdaSPEC's potential to accelerate end to end decoding in a real world setting, we use frontier inference engine vLLM [15] on one single A100 GPU and report speed-up in Table 5. Results show that an expected 10∼20% speed-up could be easily achieved compared with DistillSpec, demonstrating the effectiveness of our approach.

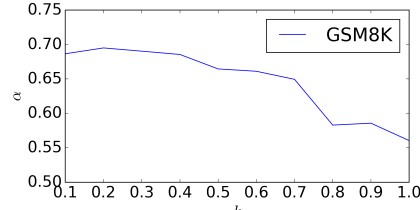

Figure 4: **Impact of token selection ratio $k$ on acceptance rate for GSM8K:** Results show a general trend where lower $k$ values (0.2-0.4) yield higher acceptance rates.

Table 5: Generation speed for AdaSPEC with VLLM on one single A100 GPU. We use greedy decoding and report time to generate a sentence and one token. We use Pythia-31M→1.4B and the models are trained for 3 epochs. On these tasks, AdaSPEC exhibits 10∼20% speed-up.

|  |  | Speed (s/sentence) | Speed (tokens/s) |
|---|---|---|---|
| MBPP | DistillSpec | 0.69 | 149.15 |
|  | AdaSPEC | **0.57** | **181.67** |
| GSM8K | DistillSpec | 0.51 | 227.86 |
|  | AdaSPEC | **0.48** | **241.34** |
| CNN/DailyMail | DistillSpec | 0.76 | 248.49 |
|  | AdaSPEC | **0.67** | **283.50** |

**Integration with Advanced SD Methods.** To demonstrate the orthogonality and generalizability of AdaSPEC beyond vanilla speculative decoding (SD), we integrate our method with EAGLE [17], an advanced SD algorithm featuring tree attention and adaptive expansion strategies. Following the standard 3-Epoch training setup on the ShareGPT dataset, we evaluate both training accuracy and generation speed on MT-Bench. As shown in Table 6, AdaSPEC consistently improves both accuracy and decoding efficiency within the EAGLE framework.

Table 6: Vicuna-7B-v1.3 [8, 31] with 3-Epoch finetuning following original EAGLE recipe. Here, training accuracy refers to first-generated-token accuracy in the training set.

|  | Eagle | Eagle + AdaSPEC |
|---|---|---|
| Training Accuracy ↑ | 75.3% | **76.3%** |
| Speed (s/sentence) ↓ | 8.85 | **8.06** (-8.9%) |
| Speed (tokens/s) ↑ | 63.48 | **68.21** (+7.45%) |

Table 7: Acceptance rate of larger model configuration with 3-Epoch GSM8K.

|  | GSM8K |
|---|---|
| DistillSpec | 84.43% |
| AdaSPEC | **86.21%** |

**Results on Larger Models.** We conduct an additional GSM8K evaluation using a combination of the Qwen2.5-0.5B and Qwen2.5-32B models. When trained with 3 epochs, AdaSPEC reaches an acceptance rate of 86.21% while DistillSpec achieves 84.43%, as shown in Table 7. This shows that our approach can easily scale up to larger models.

**Results on Mixed Dataset.** To further validate AdaSPEC's ability to work on blended tasks, we mix GSM8K with MBPP in training and validate $\alpha$ separately. Specifically, we first train the target on MBPP and then on GSM8K, each for 3 epochs. The reference and draft models also follow the same process. The

Table 8: Performance on mixed datasets.

|  | MBPP | GSM8K |
|---|---|---|
| DistillSpec | 69.63% | 72.75% |
| AdaSPEC | **70.69%** | **78.41%** |

results in Table 8 reveals that AdaSPEC strives to retain the original model's capabilities as much as possible, with less forgotten knowledge.

## 5  Dicussion

**Size Gap Between Target and Draft Models.** In traditional SD settings, the size gap between the draft model and the target model is often within 10x. In this work, we demonstrate that AdaSPEC effectively bridges the performance gap, even when the size difference is substantial — up to 64 times in our experiments. By leveraging selective token filtering and Knowledge Distillation, AdaSPEC can enhance token acceptance rates and help maintain generation quality, providing more opportunities to use significantly smaller draft models.

**Model Size Gap and Performance Gains.** From Table 1, we observe that the performance gain of AdaSPEC over DistillSpec becomes more pronounced as the size gap between the reference and target models increases (e.g., from CodeGen-350M → Phi-2 to Pythia-31M → 1.4B). This trend is consistent across both 3-Epoch and Optimal-Epoch settings. The result aligns well with our motivation: when the capacity discrepancy between models widens, direct Knowledge Distillation tends to suffer from representation mismatch, making it harder for the smaller model to absorb all teacher signals uniformly. AdaSPEC's adaptive mechanism mitigates this issue by selectively aligning easier tokens first, effectively narrowing the transfer gap. Consequently, the larger the size difference, the greater the relative improvement AdaSPEC achieves.

**Connection with Lin et al. [19].** A similar token selection method is proposed in Lin et al. [19], which focuses on identifying and prioritizing harder-to-learn tokens (opposite to the motivation of AdaSPEC) during pre-training. Different from their design, our approach focuses on addressing the limited capacity of the draft model in SD. Specifically, we focus on identifying and filtering out challenging tokens, allowing the draft model to concentrate on learning easier-to-predict tokens. Our selective distillation process ensures that the draft model aligns more effectively with the target model on tokens that are more tractable, given its constrained capacity. By doing so, we maximize the draft model's limited resources while maintaining high-quality predictions in SD tasks. Thus, the essential difference lies in the distinct objectives of pre-training and Speculative Decoding.

**Limitations.** As a preliminary study on selective training for SD, we limit our study on simple loss-related token filter. In future work, one can design more adaptive filtering strategies as well as integrate AdaSPEC with tree-based or multi-step verification frameworks to further improve both speed and quality of LLM inference.

## 6  Conclusion

We present AdaSPEC, a novel approach for training more efficient draft models for SD. AdaSPEC introduces selective token filtering based on reference model perplexity gaps, enabling draft models to focus limited capacity on tokens where alignment with the target model is most achievable. Experiments show it outperforms baselines in arithmetic reasoning, instruction following, code generation, and summarization with higher acceptance rates.

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

# A Technical Appendices and Supplementary Material

## A.1 Full Algorithms for AdaSPEC

---

**Algorithm 1** Greedy Speculative Decoding

---

1: **Input:** target model $M_p$, draft model $M_q$, input sequence $\boldsymbol{x}$
2: $\text{accept} \leftarrow 0, \text{reject} \leftarrow 0, t \leftarrow \text{len}(\boldsymbol{x})$
3: ▷ Sample $\gamma$ tokens $z_1, ..., z_\gamma$ from $M_q$ autoregressively.
4: **for** $i = 1$ to $\gamma$ **do**
5: $\quad z_i = S_q(\boldsymbol{z}_{<i}); \boldsymbol{z} \leftarrow \boldsymbol{z} + z_i$
6: **end for**
7: ▷ Verify in parallel
8: $\boldsymbol{z}' \leftarrow [S_p(\boldsymbol{z}_{<1}), S_p(\boldsymbol{z}_{<2}), ..., S_p(\boldsymbol{z}_{<\gamma})] \triangleq [z_1', z_2', ..., z_\gamma']$
9: **for** $i = 1$ to $\gamma$ **do**
10: $\quad$ **if** $z_i' \neq z_i$ **then**
11: $\quad\quad \text{reject} \leftarrow \text{reject} + 1$; **break**
12: $\quad$ **end if**
13: $\quad \boldsymbol{x} \leftarrow \boldsymbol{x} + z_i, \text{accept} \leftarrow \text{accept} + 1$
14: $\quad$ **if** $z_i = \text{<eos>}$ **then**
15: $\quad\quad$ **return** $\boldsymbol{x}, \text{accept}, \text{reject}$
16: $\quad$ **end if**
17: **end for**
18: $\boldsymbol{x} \leftarrow \boldsymbol{x} + S_p(\boldsymbol{z}_{<1})$; **goto** 4

---

---

**Algorithm 2** AdaSPEC: Selective Distillation for Speculative Decoding

---

1: **Input:** dataset $\mathcal{D}$, target model $M_p$, draft model $M_q$, fraction $k$
2: **Step 1. Fine-tune $M_p$:**
3: $\quad$ Train $M_p$ on $\mathcal{D}$ (via standard LM fine-tuning) to obtain $M_p^*$.
4: **Step 2. Reference Model Training:**
5: $\quad$ Initialize $M_{\text{ref}} \leftarrow M_q$.
6: $\quad$ Distill $M_{\text{ref}}$ from $M_p^*$ on $\mathcal{D}$ (e.g. forward KL).
7: **Step 3. Selective Distillation:**
8: $\quad$ *(a) Compute losses:* For each token $w$ and context,

$$\mathcal{L}_{\text{ref}}(w) = \mathcal{K}\left(P(w \mid \text{context}) || R(w \mid \text{context})\right),$$
$$\mathcal{L}_{\text{draft}}(w) = \mathcal{K}\left(P(w \mid \text{context}) || Q(w \mid \text{context})\right),$$
$$\Delta_{\mathcal{L}}(w) = \mathcal{L}_{\text{draft}}(w) - \mathcal{L}_{\text{ref}}(w).$$

9: $\quad$ *(b) Filter tokens:*

$$\mathbb{S} = \left\{ w \mid \Delta_{\mathcal{L}}(w) \text{ is among the top } k \times 100\% \text{ of all tokens} \right\}, \quad k \in [0, 1].$$

10: $\quad$ *(c) Distill on filtered set:*

$$\min_{M_q} \mathcal{L}_{\text{distill}} = \frac{1}{k \cdot |\boldsymbol{y}|} \sum_{i=1}^{|\boldsymbol{y}|} \mathbb{I}\left[y_i \in \mathbb{S}\right] \cdot \mathcal{L}_{\text{draft}}(y_i)$$

11: **return** $M_q$ (draft model in SD)

---

## A.2 Implementation Details

We use the hyperparameters in Table 9. For 3-Epoch setting, both reference and draft model are distilled for 3 epochs. For Optimal-Epoch setting, the target model is first fine-tuned to maximize performance on validation set. Specifically, for GSM8K, the number of target epochs is chosen

according to validation accuracy, while for the rest of the experiments it is chosen according to validation perplexity. Afterwards, we distill the reference model and pick the one with highest $\alpha$ on validation set. Eventually, this model serves as the reference model to train our draft model. For robustness, we only select the optimal epoch from 1, 3, 6, 10, 15, 20 and 30 (for XSUM and CNN/Daily Mail we select from 1, 3, 6, 10 for training efficiency). Note that when performing token selection, we apply the linear scaling rule for learning rate adjustment [12].

Table 9: Experimental hyperparameters.

| Task | Hyperparameter | 3-Epoch | | Optimal-Epoch | |
|---|---|---|---|---|---|
| | | Pythia 31M→1.4B | Codegen-350M→Phi-2 | Pythia 31M→1.4B | Codegen-350M→Phi-2 |
| GSM8K | Batch size | 16 | 16 | 16 | 16 |
| | Learning rate | 3e-4 | 3e-4 | 3e-4 | 3e-4 |
| | Epochs for target model | 3 | 3 | 6 | 3 |
| | Epochs for reference model | 3 | 3 | 15 | 30 |
| | Epochs for draft model | 3 | 3 | 30 | 30 |
| | Filter fraction $k$ | 0.4 | 0.4 | 0.4 | 0.4 |
| Alpaca | Batch size | 16 | 16 | 16 | 16 |
| | Learning rate | 3e-4 | 3e-4 | 3e-4 | 3e-4 |
| | Epochs for target model | 3 | 3 | 1 | 1 |
| | Epochs for reference model | 3 | 3 | 15 | 20 |
| | Epochs for draft model | 3 | 3 | 30 | 15 |
| | Filter fraction $k$ | 0.4 | 0.4 | 0.4 | 0.4 |
| MBPP | Batch size | 8 | 8 | 8 | 8 |
| | Learning rate | 1e-5 | 1e-4 | 1e-5 | 1e-4 |
| | Epochs for target model | 3 | 3 | 1 | 1 |
| | Epochs for reference model | 3 | 3 | 30 | 10 |
| | Epochs for draft model | 3 | 3 | 6 | 6 |
| | Filter fraction $k$ | 0.4 | 0.4 | 0.4 | 0.4 |
| CNN/Daily Mail | Batch size | 16 | 16 | 16 | 16 |
| | Learning rate | 1e-4 | 1e-4 | 1e-4 | 1e-4 |
| | Epochs for target model | 3 | 3 | 1 | 1 |
| | Epochs for reference model | 3 | 3 | 10 | 10 |
| | Epochs for draft model | 3 | 3 | 10 | 10 |
| | Filter fraction $k$ | 0.4 | 0.4 | 0.4 | 0.4 |
| XSUM | Batch size | 16 | 16 | 16 | 16 |
| | Learning rate | 3e-4 | 1e-4 | 3e-4 | 1e-4 |
| | Epochs for target model | 3 | 3 | 1 | 1 |
| | Epochs for reference model | 3 | 3 | 10 | 10 |
| | Epochs for draft model | 3 | 3 | 10 | 10 |
| | Filter fraction $k$ | 0.4 | 0.4 | 0.4 | 0.4 |

### A.3 AdaSPEC Example Tokens

Here, we showcase some example tokens (Listing 1) that AdaSPEC selects while training on GSM8K. These selected tokens are typically mathematical related tokens, such as digits and operators.

```
{ "scored", "8", "in", "thus", "9", "x", "1", "=", "<<", "9", "*", "91", "=", "19",
    ">>", "8", "19", "Em", "because", "28", "28", "\+", "8", "28", "+", "90", "18",
     "9", "18", "18", "18", "-", "8", "=", "99", "99", "99", "+", "100", "The", "
    final", "answer", "100", "equal", "12", "+", "7", "=", "19", ">>", "19", "packs
    ", "19", "5", "24", "total", "(", "24", "*(", "2", ")=", "16", ">>", "16", "J",
     "spends", "inside", "because", "-", "(", "inside", "16", "iley", "3", "18", "
    spends", "12", "In", "total", "they", "+", "12", "=", "<<", "8", "+", "=", "20",
     "/", "=", "10", "10", "The", "earned", "final", "answer", "difference", "-",
    "=", "13", "*", "2", "26", ">>", "26", "twice", "26", "18", "=", "26", "18",
    "8", "The", "final", "answer", ":", " ", "8", }
```

Listing 1: Selected tokens during GSM8k training.

### A.4 Minimal Code Implementation of AdaSPEC

The core of AdaSPEC could be implemented with $\sim$ 100 lines of code. We show it in Listing 2.

```
def compute_loss(
        self,
        model,
        inputs,
        return_outputs=False,
```

```
        num_items_in_batch=None
):
    labels = inputs["labels"][:, 1:]

    outputs = model(**inputs)
    with torch.no_grad():
        target_outputs = self.target_model(**inputs)
        ref_outputs = self.ref_model(**inputs)

    logits = outputs["logits"]
    target_logits = target_outputs["logits"]
    ref_logits = ref_outputs["logits"]

    loss_fct = KLDivLoss(reduction="none")

    shift_logits = logits[..., :-1, :].contiguous()
    shift_target_logits = (target_logits[..., :-1, :]
                            .contiguous())
    shift_ref_logits = (ref_logits[..., :-1, :]
                        .contiguous())

    shift_logits = shift_logits.view(
        -1, model.config.vocab_size)
    shift_target_logits = (shift_target_logits
                            .view(-1, model.config.vocab_size))
    shift_ref_logits = (shift_ref_logits
                        .view(-1, model.config.vocab_size))
    mask = labels.ne(IGNORE_INDEX).flatten().unsqueeze(-1)

    shift_logits = (
        torch.masked_select(shift_logits, mask=mask)
        .view(-1, model.config.vocab_size))
    shift_target_logits = \
        (torch.masked_select(shift_target_logits, mask=mask)
         .view(-1, model.config.vocab_size))
    shift_ref_logits = \
        (torch.masked_select(shift_ref_logits, mask=mask)
         .view(-1, model.config.vocab_size))

    p = F.softmax(shift_target_logits, dim=-1)
    q_log = F.log_softmax(shift_logits, dim=-1)
    actual = loss_fct(q_log, p)

    q_log = F.log_softmax(shift_ref_logits, dim=-1)
    ref = loss_fct(q_log, p)

    actual = actual.sum(dim=-1)
    ref = ref.sum(dim=-1)

    k = self.k
    delta = actual - ref
    mask = delta >= torch.quantile(
        delta, 1 - k, dim=0, keepdim=True)

    if num_items_in_batch is not None:
        loss = torch.masked_select(actual, mask=mask).sum()
        loss = loss / num_items_in_batch
    else:
        loss = torch.masked_select(actual, mask=mask).mean()

    return (loss, outputs) if return_outputs else loss
```

Listing 2: AdaSPEC implementation with PyTorch.

### A.5  Broader Impact

AdaSPEC can be used mainly to improve generation speed of LLMs, which is a positive influence to reduce potential electric energy consumption for serving LLMs. However, this technique may also be used for some models that is non-compliance with regulations and ethics, such as models that generate discriminatory contents.

### A.6  Experiments compute resources

Here we list the estimated GPU hours; see Table 10.

Table 10: GPU hours of training models on A100 GPUs.

| Task | 3-Epoch | | Optimal-Epoch | |
|---|---|---|---|---|
| | Pythia-31M → 1.4B | CodeGen-350M → Phi-2 | Pythia-31M → 1.4B | CodeGen-350M → Phi-2 |
| GSM8K | 1 | 3 | 15 | 50 |
| Alpaca | 1 | 3 | 15 | 50 |
| MBPP | 1 | 3 | 15 | 50 |
| CNN/Daily Mail | 60 | 200 | 200 | 700 |
| XSUM | 60 | 200 | 200 | 700 |

