# OpenReview forum: "AdaSPEC: Selective Knowledge Distillation for Efficient Speculative Decoders"
_NeurIPS.cc/2025/Conference — NeurIPS 2025 spotlight_

### Official Review · Reviewer_5tif · 2025-06-09

**Clarity:** 4
**Significance:** 3
**Originality:** 3
**Rating:** 4
**Confidence:** 5

**Summary:**

This paper introduces AdaSPEC, a novel method to improve the efficiency of Speculative Decoding (SD).
AdaSPEC addresses the problem of the small draft model struggling to fully learn the target knowledge by incorporating selective token filtering into the Knowledge Distillation (KD) process, using a reference model to identify and exclude diffucult tokens.
By focusing distillation on simpler tokens, AdaSPEC enables the draft model to better align with the target model, leading to higher token acceptance rates and improved inference speed without compromising generation quality.

**Questions:**

1. Can AdaSPEC generalize beyond vanilla SD to more advanced methods like Eagle or Medusa?

2. Could more relevant baselines be included, like Rho-1 mentioned in the paper?

3. Considering the significant gap between 3-epoch and optimal-epoch performance, why wasn’t a more effective hyperparameter selection strategy proposed to improve results?

**Ethical Concerns:**

["NO or VERY MINOR ethics concerns only"]

**Final Justification:**

The additional experiments during rebuttal (on SOTA speculative decoding methods like Eagle and Medusa) prove that adaspec can generalize to arbitrary speculative decoding methods. Hence, my main concerns are addressed.

This work is an orthogonal work to most of the existing speculative decoding methods, targeting at the pruning difficult tokens to improve acceptance rate. I appreciate the novelty and significance of the proposed method, and believe it will be of interest to a wider range of audience.

**Limitations:**

yes

**Quality:**

3

**Strengths And Weaknesses:**

Strengths:
1. AdaSPEC is simple to implement and is a plug-and-play method.
2. AdaSPEC achieves much higher acceptance rate than DistillSPEC baseline.

Weaknesses:
1. Only the vanilla SD is tested, so it is not clear whether AdaSPEC could generalize to more advanced SD methods like Eagle.
2. The only baseline is DistillSPEC. The evaluation is limited in scope. The discussion of related work is limited.
3. There is a huge gap between 3-epoch and optimal-epoch performance. A more advanced hyper-parameter selection strategy could have been proposed to optimize the performance.

---

> ### Author Rebuttal · Authors · 2025-07-30
>
> Dear Reviewer 5tif,
>
> Thank you for recognizing the potential and effectiveness of our work and for providing detailed constructive comments. Below, we address each point raised.
>
> ---
>
> >**W1**: Only the vanilla SD is tested, so it is not clear whether AdaSPEC could generalize to more advanced SD methods like Eagle.
>
> **R1**: Thank you for your insightful comment. To address this, we implemented AdaSPEC with Eagle—a representative SD method featuring tree attention that demonstrates superior performance to Medusa. Following the standard practice in the Eagle paper [1], we trained the Eagle decoding head on the ShareGPT dataset using the 3-epoch setting from our work and evaluated generation speed on MT-bench. The results below demonstrate AdaSPEC's effectiveness and generalizability beyond vanilla SD to more advanced SD methods like Eagle.
>
> *Table R1.* Vicuna v1.3 7B with 3-epoch ShareGPT. Here, train accuracy refers to first-generated-token accuracy in the training set.
>
> |                                 | **Eagle** | **Eagle + AdaSpec** |
> | ------------------------------- | --------- | ------------------- |
> | Train ACC ↑                     | 75.3%     | **76.3%**           |
> | Generation Speed (s/sentence) ↓ | 8.85      | **8.06** (-8.9%)    |
> | Generation Speed (tokens/s) ↑   | 63.48     | **68.21** (+7.45%） |
>
> ---
>
> >**W2**: The only baseline is DistillSPEC. The evaluation is limited in scope. The discussion of related work is limited.
>
> **R2**: We appreciate the reviewer’s feedback and the opportunity to clarify. As of now, DistillSpec is the only existing work that applies model distillation to train the draft model in the speculative decoding (SD) framework, and it is widely regarded as the state-of-the-art method in this line of research. Therefore, we believe our comparison to DistillSpec is both appropriate and meaningful within the scope of distillation-based SD methods.
>
> Moreover, our results demonstrate not only superior decoding efficiency under the standard SD architecture but also stronger generalization across models of different sizes. We have also highlighted that **our approach is orthogonal to other classes of SD techniques** (e.g.adaptive rejection, or sampling strategies), and can in principle be combined with them to further enhance performance. For example, our experimental results in R1 above demonstrate that our methods can be effectively integrated into the EAGLE framework.
>
> Regarding the discussion of related work, we have included comparisons to both speculative decoding and distillation literature in Section 2. We agree that additional connections—such as to broader acceleration strategies—may be further explored in future iterations of this work.
>
> ---
>
> >**W3**: There is a huge gap between 3-epoch and optimal-epoch performance. A more advanced hyper-parameter selection strategy could have been proposed to optimize the performance.
>
> **R3**: Please refer to Q3 for this.
>
> ---
>
> >**Q1**: Can AdaSPEC generalize beyond vanilla SD to more advanced methods like Eagle or Medusa?
>
> **R1**: Yes. Please refer to W1 for the corresponding results.
>
> ---
>
> >**Q2**: Could more relevant baselines be included, like Rho-1 mentioned in the paper?
>
> **R2**: Though AdaSpec resembles Rho-1 in the token selection stage, the two studies are different. Specifically, Rho-1 focuses on the pretraining of language models, which is not relevant to our study. We have no idea how AdaSpec could be compared with Rho-1.
>
> ---
>
> >**Q3**: Considering the significant gap between 3-epoch and optimal-epoch performance, why wasn’t a more effective hyperparameter selection strategy proposed to improve results?
>
> **R3**: First, we have already tuned our hyperparameters to their optimal as much as possible within each setting. The hyperparameters can be found in the appendix.
>
> Importantly, the goal of our work is not to maximize absolute performance via exhaustive hyperparameter tuning, but rather to evaluate and compare the effectiveness of AdaSpec across different realistic training scenarios. As discussed in Section 4.2, we deliberately introduce two representative training settings to simulate common use cases:
>
> The **3-epoch setting** simulates a realistic case where the draft model has been *partially trained* and already performs reasonably well on the target domain. This reflects a typical ablation setup, where forgetting is limited and generalization is already strong.
>
>
>
> The **optimal-epoch setting** aims to approximate the *best achievable performance* on the target domain with further training, serving as an upper bound for model capability.
>
> In addition, the **3-epoch setting also aligns with our mixed-task experiments in Section 4.6**, where we train models sequentially on MBPP and then GSM8K (each for 3 epochs) to test AdaSpec’s ability to retain previously acquired knowledge. As shown in Table 6, AdaSpec is effective in preserving the model’s original capabilities under such limited training regimes, further supporting the design choice of the 3-epoch setting.
>
> Finally, we note that AdaSpec consistently improves performance under both settings, demonstrating strong generality. Importantly, these gains are achieved **without task-specific hyperparameter tuning**, highlighting the robustness and practicality of our approach.
>
> [1] EAGLE: Speculative Sampling Requires Rethinking Feature Uncertainty

---

> > ### Comment · Reviewer_5tif · 2025-08-04
> >
> > Thank you for your response. Please include the relevant results in the future version. I believe it will improve the paper's quality. I will increase my score to 4.

---

> ### Author Response · Authors · 2025-08-04
> **Thank you for raising the score!**
>
> Thank you again for providing detailed constructive comments and for raising the score! We will update the experiment results in subsequent version.

---

### Official Review · Reviewer_qrev · 2025-06-30

**Clarity:** 4
**Significance:** 3
**Originality:** 4
**Rating:** 5
**Confidence:** 4

**Summary:**

This paper introduces AdaSPEC, an adaptive loss for model distillation for later use in Speculative Decoding settings. Based on the observation that a focus on simpler tokens should enhance the acceptance ratio during SD, the work proposes a two-step distillation, the first step being a normal KL-based distillation into a smaller reference model and during a second stage, another KL-based distillation of the final draft model on selected "simple" tokens derived by the KL-difference between reference and draft models. This approach can consistently improve acceptance scores over the DistillSPEC baseline.

**Questions:**

- Is there a possibility to train both reference and draft model at the same time? This would bring down the large cost of evaluating the big target model twice.

- Can you explain the only mild improvements of acceptance scores (often ~1%)?

- Could you add in your discussion of SD alternatives like MoE, CoT and recurrent-depth approaches in l.25ff? What is your opinion on those compared to SD?

**Ethical Concerns:**

["NO or VERY MINOR ethics concerns only"]

**Final Justification:**

I want to recommend this paper for acceptance as it provides an elegant approach to improve acceptance scores of speculative decoding, accompanied with sound experimentation.

**Limitations:**

Within the narrow focus of this paper (Speculative Decoding setting), it addresses the main limitations.

**Paper Formatting Concerns:**

None.

**Quality:**

3

**Strengths And Weaknesses:**

Strengths:
- Improved acceptance scores for Speculative Decoding

Weaknesses:
- Twice the distillation compute
- Limited to Speculative Decoding settings

---

> ### Author Rebuttal · Authors · 2025-07-30
>
> Dear Reviewer qrev,
>
> Thank you for recognizing the potential and effectiveness of our work and for providing detailed constructive comments. Below, we address each point raised.
>
> ---
>
> >**W1**: Twice the distillation compute
>
> **R1**: While it is true that our approach introduces additional compute during the distillation stage, the core objective of speculative decoding is to accelerate inference. We argue that a moderate increase in training cost is acceptable, especially given the substantial inference speedups achieved. Moreover, compared to pretraining large language models, the overall cost of distillation on downstream tasks is relatively small and typically affordable in practice.
>
> ---
>
> >**W2**: Limited to Speculative Decoding settings
>
> **R2**: Our method targets the standard speculative decoding (SD) setting, which is a widely adopted and practically relevant scenario for accelerating inference in large language models. In this context, our approach achieves state-of-the-art performance and outperforms DistillSpec in both efficiency and generalization.
>
> To further validate AdaSpec on more advanced approaches beyond vanilla SD, we have now included additional experiments combining our method with Eagle, a popular decoding strategy utilizing tree attention for multi-token speculation. Following the standard practice in the Eagle paper [1], we trained the Eagle decoding head on the ShareGPT dataset using the 3-epoch setting from our work and evaluated generation speed on MT-bench.
>
> *Table R1.* Vicuna v1.3 7B with 3-epoch ShareGPT. Here, train accuracy refers to first-generated-token accuracy in the training set.
>
> |                                 | **Eagle** | **Eagle + AdaSpec** |
> | ------------------------------- | --------- | ------------------- |
> | Train ACC ↑                     | 75.3%     | **76.3%**           |
> | Generation Speed (s/sentence) ↓ | 8.85      | **8.06** (-8.9%)    |
> | Generation Speed (tokens/s) ↑   | 63.48     | **68.21** (+7.45%） |
>
> The results in the table above demonstrate AdaSPEC's effectiveness and generalizability beyond vanilla SD to more advanced SD methods like Eagle.
>
> Could you please clarify which specific scenarios are considered missing? We believe the standard SD setting and the additional experiment above already provides a strong and representative benchmark for evaluating speculative decoding methods.
>
> ---
>
> >**Q1**: Is there a possibility to train both reference and draft model at the same time?
>
> **R1**: Yes. Although the reference model is limited to a roughly pre-finetuned speculative decoder in our study, it can be replaced by any small model that aligns with the target model. One possible approach to train both reference and draft model simultaneously is to dynamically assign the reference model to be an earlier checkpoint of the draft model. We find this idea quite interesting and believe it opens up a new direction for adaptive speculative decoding, which we plan to explore in future work.
>
> ---
>
> >**Q2**: Can you explain the only mild improvements of acceptance scores (often ~1%)?
>
> **R2**: The reason for modest improvement in our main results is mainly due to the small size gap between draft and target model. In Pythia settings, where the draft/target size ratio is about 1/40, AdaSpec introduces up to 8% acceptance rate improvement. When it comes to some of the Phi settings, whose draft/target ratio is about 1/8, that advantage is weakened to up to 3%. This shows the advantage of our method on extreme draft/target ratios: the larger the size gap is, the more effective our method is.
>
> ---
>
> >**Q3**: Could you add in your discussion of SD alternatives like MoE, CoT and recurrent-depth approaches in l.25ff? What is your opinion on those compared to SD?
>
> **R3**: We thank the reviewer for the suggestion. However, we would appreciate further clarification on the intended comparison, as the listed approaches seem to target different aspects of model design and usage.
>
> To our understanding:
>
> - **MoE (Mixture of Experts)** accelerates inference by activating only a subset of model parameters during each forward pass, providing efficiency gains through activation sparsity. In contrast, speculative decoding is model-agnostic and focuses on speeding up generation by parallelizing the decoding process.
>
>
>
> - **Chain-of-Thought (CoT)** and **recurrent-depth** methods aim to improve reasoning capabilities or model expressivity, often at the cost of dynamically expanding inference computation. These approaches are typically used to enhance output quality rather than inference efficiency.
>
> [1] EAGLE: Speculative Sampling Requires Rethinking Feature Uncertainty

---

> > ### Comment · Reviewer_qrev · 2025-08-02
> >
> > Thank you for your clarifications.
> >
> > The second weaknesses merely reflects the sharp setting of the method but should not be seen as a major downside. I want to highlight that within this focus, the work presents an important contribution and is technically sound, so I want to recommend it for acceptance.
> >
> > Regarding Q3, my main intention was to motivate extending the discussion on inference optimization techniques for other methods on adaptive computation. This would put speculative decoding into a broader context of approaches, maybe even highlighting why speculative decoding is a better choice over others.

---

> ### Author Response · Authors · 2025-08-04
> **Thank you for the valuable review!**
>
> Thank you again for the valuable review! We really appreciate your recommendation for acceptance, and we would like to make more clarifications regarding Q3, as well as in the subsequent version of the paper.
>
> Compared to MoE, CoT, and recurrent-depth approaches, the biggest advantage of Speculative Decoding is lossless acceleration—that is, the expected accuracy remains entirely consistent with the vanilla counterpart. SD can achieve over a 2x speed improvement with just a little extra memory for inference.
>
> Compared with SD, MoE has always lagged behind its dense counterpart in performance (mainly due to convergence issues difficulties introduced by sparsity). Besides, the CoT paradigm does not consistently enhance model reasoning capabilities, as pointed out by some recent studies such as [2].
>
> Our method demonstrates that through appropriate token level pruning, the performance of SD can be further improved on a lossless basis, offering a simple and effective new approach.
>
> [2] To CoT or not to CoT? Chain-of-thought helps mainly on math and symbolic reasoning

---

### Official Review · Reviewer_TYdk · 2025-07-01

**Clarity:** 3
**Significance:** 3
**Originality:** 3
**Rating:** 5
**Confidence:** 4

**Summary:**

This paper examines the implications of the observation that Speculative Decoding should prioritize the accuracy of predicted tokens over optimizing the overall KL divergence between the target and the reference model. Consequently, it proposes AdaSPEC, which prioritizes distillation of the draft model on the subset of tokens that it is actually able to learn to predict properly, given its small scale. The heuristic used by AdaSPEC to choose this subset of tokens is to select those where the most learning progress has been made so far in terms of KL divergence with respect to the target model. Results show a modest but consistent gain in acceptance rate compared to the ablation of the selective-token filtering (corresponding to DistillSpec \[1\]).

\[1\] Zhou, Y., Lyu, K., Rawat, A. S., Menon, A. K., Rostamizadeh, A., Kumar, S., Kagy, J., & Agarwal, R. (2023). DistillSpec: Improving Speculative Decoding via Knowledge Distillation. *ArXiv*. [https://arxiv.org/abs/2310.08461](https://arxiv.org/abs/2310.08461)

**Questions:**

* In L138, you state that the selected tokens are the ones that are easiest for the draft model to align with the target model. However, this motivation is slightly unclear to me. In particular, it seems that tokens maximizing (9) might satisfy a combination of “learnability”, related to your “ease”, but also “not yet learnt”, since tokens that are already predicted by the reference model well will be easy to predict, but will not satisfy (9). Would you agree?
* In L182, could you mention in the appendix and reference how you specifically chose the number of epochs?
* In L188-189, for reproducibility, it would be important to report how the hyperparameters and most importantly, the filter fraction were chosen. In particular, could you justify in L286 why you sometimes use k=0.8 instead of 0.4 everywhere? Is there any way to understand when we would want a higher k and when a lower one?
* In L198, could you please provide the respective plots for the other configurations in the appendix, given that these should be readily available already? That would aid in the completeness of the results.

### Other suggestions

* In the equation after L140, you mention selecting the top k tokens, but in the appendix pseudo-code, you appear to choose them based on quantiles instead. I would encourage addressing this inconsistency for reproducibility.
* I would encourage the authors in the Abstract and Introduction to focus more on the observation that simple KL optimization does not effectively maximize the acceptance rate, which is arguably the goal of SD. Therefore, in SD, we aim for high precision or maximum likelihood, not matching the distribution. As you rightly mention in the limitations, your heuristic is just a first approach to use this observation. Still, you could capitalize on this observation as a key contribution, which might enhance the impact of this work.

**Ethical Concerns:**

["NO or VERY MINOR ethics concerns only"]

**Final Justification:**

After reconsidering the paper in light of the other reviews, the author’s rebuttals, and the additional experiments. I choose to maintain my original score.

Nonetheless, I would encourage the authors to strengthen their related work discussion if the paper is accepted, potentially using an additional page or an appendix to discuss the broader literature in speculative decoding and how their approach is orthogonal to some of the approaches, and for what methods it might not be readily applicable.

I would also appreciate a more specific measurement or an in-depth discussion of the particular impact of the additional inference step compared to the vanilla cost in the paper, even if the focus of SD is on the inference cost. I believe it would be of interest to practitioners and for comparison with future works.

**Limitations:**

yes

**Paper Formatting Concerns:**

No major formatting issues were found.

#### Minor

* The green lines in Alg 1 are somewhat hard to read.
* In Listing 2, I would appreciate it if the code were smaller and properly formatted for improved readability.

**Quality:**

3

**Strengths And Weaknesses:**

## Strengths

* **Clarity.** I find the paper quite well-written and clear. The authors effectively motivate the problem and clearly explain their approach, and the paper’s structure is logical.
* **Problem significance**. The problem of increasing the computational efficiency of LLMs is very relevant and timely, and the paper’s results, although modest, could have a huge impact on the large scales of both usage and cost of LLM inference.
* **Originality**. Although token selection by difficulty \[2\], as discussed by the authors, or the hardness of examples in general \[3\] is not novel, the reason why this paper considers difficulty comes from a different perspective and has different implications. The observation that leads the author to adaptive token selection by difficulty, in my opinion, is where the originality of this paper lies, and it opens an avenue for future work to examine different designs to achieve a higher acceptance rate, beyond simply considering token difficulty.
* **Quality/Evaluation**. Evaluation is overall quite diverse, both in terms of architectures and datasets. Together with the analysis and ablation study experiments, the empirical results appear robust and sufficient \[See questions below for further details\].

## Weaknesses

* **Modest empirical results.** Although consistent, the effect of the method is modest, which reduces the impact of the work.
* **Computation cost.** Although the implementation of token filtering is simple and should not incur a significant computational or memory cost, given that it involves the additional inference of the reference model, I would appreciate a comparison of the impact, given an equal number of tokens incorporated in the GD step.
* **Reproducibility details.** Some minor implementation choices could be stated more clearly, see the questions below.

---

The main limitation that stops me from increasing the score is the modesty of the empirical results, which I am unsure if this can be addressed during the rebuttal. However, I would greatly encourage the authors to consider and incorporate the other comments.

\[2\] Lin, Z., Gou, Z., Gong, Y., Liu, X., Shen, Y., Xu, R., Lin, C., Yang, Y., Jiao, J., Duan, N., & Chen, W. (2024). Rho-1: Not All Tokens Are What You Need. *ArXiv*. [https://arxiv.org/abs/2404.07965](https://arxiv.org/abs/2404.07965)
\[3\] Seedat, N., Crabbé, J., & Bica, I. (2022). Data-IQ: Characterizing subgroups with heterogeneous outcomes in tabular data. *ArXiv*. [https://arxiv.org/abs/2210.13043](https://arxiv.org/abs/2210.13043)

---

> ### Author Rebuttal · Authors · 2025-07-30
>
> Dear Reviewer TYdk,
>
> Thank you for recognizing the potential and effectiveness of our work and for providing detailed constructive comments. Below, we address each point raised.
>
> ---
>
> >**W1**: Although consistent, the effect of the method is modest, which reduces the impact of the work.
>
> **R1**: The reason about modest improvement in our main results is mainly due to small size gap between draft and target model. In Pythia settings, where the draft/target size ratio is about 1/40, AdaSpec introduces up to 8% accuracy improvement. When it comes to the Phi settings, whose draft/target ratio is about 1/8, that advantage is weakened to up to 3%. This shows the advantage of our method on extreme draft/target ratios: the larger the size gap is, the more effective our method is.
>
> Furthermore, we would like to highlight that the advantage of AdaSpec is in inference acceleration but not training. Although AdaSpec introduces **minimal overhead to the standard distillation framework**, it is actually very effective in **inference-heavy scenarios**. Compared with the marginal computation cost of distillation, AdaSpec can lead to **10–20% speedup** (as shown in Section 4.6), making it highly practical for real-world deployment.
>
> Finally, to further enhance the practical utility and impact of our work, we integrate AdaSpec to Eagle, a frontier method that combines tree attention with SD. Following the standard practice in the Eagle paper [1], we trained the Eagle decoding head on the ShareGPT dataset using the 3-epoch setting from our work and evaluated generation speed on MT-bench. The results below demonstrate AdaSPEC's effectiveness and generalizability beyond vanilla SD to more advanced SD methods like Eagle.
>
> *Table R1.* Vicuna v1.3 7B with 3-epoch ShareGPT. Here, train accuracy refers to first-generated-token accuracy in the training set.
>
> |                                 | **Eagle** | **Eagle + AdaSpec** |
> | ------------------------------- | --------- | ------------------- |
> | Train ACC ↑                     | 75.3%     | **76.3%**           |
> | Generation Speed (s/sentence) ↓ | 8.85      | **8.06** (-8.9%)    |
> | Generation Speed (tokens/s) ↑   | 63.48     | **68.21** (+7.45%） |
>
> These results demonstrate that AdaSpec is not only scalable and lightweight, but also delivers **substantial practical gains** in scenarios where inference cost dominates—underscoring its relevance for efficiency-critical applications.
>
> ---
>
> >**W2**: Although the implementation of token filtering is simple and should not incur a significant computational or memory cost, given that it involves the additional inference of the reference model, I would appreciate a comparison of the impact, given an equal number of tokens incorporated in the GD step.
>
> **R2**: We argue that the extra computation cost would be marginal and can be ignored.
>
> 1. The extra computational cost mainly involves one forward pass of the reference model. Since the reference model and draft model share the same architecture and size, this would introduce about 25% extra computation (the backward pass of the draft model is twice as much as the forward pass, and gradient checkpointing introduces an extra forward pass).
> 2. In distillation we also need to run the target forward pass, whose computation is tens of the forward pass as the draft model.
>
> Most importantly, the **core goal of speculative decoding is to accelerate inference**, not to reduce training cost. Our method offers substantial decoding speedup during inference with only a small, one-time overhead during distillation training. We believe this trade-off is both reasonable and practical.
>
> ---
>
> >**Q1**: In L138, you state that the selected tokens are the ones that are easiest for the draft model to align with the target model. However, this motivation is slightly unclear to me. In particular, it seems that tokens maximizing (9) might satisfy a combination of “learnability”, related to your “ease”, but also “not yet learnt”, since tokens that are already predicted by the reference model well will be easy to predict, but will not satisfy (9). Would you agree?
>
> **R1**: Thank you for raising the question. Your understanding is correct. We will make this clear with more explanation at L138.
>
> ---
>
> >**Q2**: In L182, could you mention in the appendix and reference how you specifically chose the number of epochs?
>
> **R2**: For GSM8K, the number is chosen according to accuracy on validation set. For the rest of the settings, that number is chosen according to Cross Entropy loss on validation set.
>
> ---
>
> >**Q3**: In L188-189, for reproducibility, it would be important to report how the hyperparameters and most importantly, the filter fraction were chosen. In particular, could you justify in L286 why you sometimes use k=0.8 instead of 0.4 everywhere? Is there any way to understand when we would want a higher k and when a lower one?
>
> **R3**: We naturally choose some batch size and learning rate that appears reasonable. Alpaca, XSUM and CNN/DailyMail datasets are relatively large, so we modify some of the settings accordingly. We further tune the epochs to get optimal results.
>
> For Alpaca, we first train a set of ablations including different k, learning rate and batch size. Then we select the optimal result which appears to be 0.8. For the rest of the experiments, since 0.4 is a feasible choice for most of the settings, we keep k=0.4 and no longer tune this parameter.
>
> ---
>
> >**Q4**: In L198, could you please provide the respective plots for the other configurations in the appendix, given that these should be readily available already? That would aid in the completeness of the results.
>
> **R4**: Yes, we will provide more plots. Since NeurIPS 2025 forbids plots in author reviewer discussion, we will update this in the final version of our paper.
>
> ---
>
> >**S1**: In the equation after L140, you mention selecting the top k tokens, but in the appendix pseudo-code, you appear to choose them based on quantiles instead. I would encourage addressing this inconsistency for reproducibility.
>
> **R1**: Sorry for the misunderstanding. The "top k tokens" refers to a ratio of the selected tokens, for example, top 40% tokens. Here, k $\in$ [0,1]. We will clarify this in the paper.
>
> ---
>
> >**S2**: I would encourage the authors in the Abstract and Introduction to focus more on the observation that simple KL optimization does not effectively maximize the acceptance rate, which is arguably the goal of SD. Therefore, in SD, we aim for high precision or maximum likelihood, not matching the distribution. As you rightly mention in the limitations, your heuristic is just a first approach to use this observation. Still, you could capitalize on this observation as a key contribution, which might enhance the impact of this work.
>
> **R2**: Thank you for pointing this out. We will emphasize this insight in the introduction and abstract sections.
>
> [1] EAGLE: Speculative Sampling Requires Rethinking Feature Uncertainty

---

> > ### Comment · Reviewer_TYdk · 2025-08-01
> >
> > I appreciate the author’s rebuttal comments, which address my remarks.
> >
> > After reconsidering the paper in light of the other reviews, the author’s rebuttals, and the additional experiments. I choose to maintain my original score.
> >
> > Nonetheless, I would encourage the authors to strengthen their related work discussion if the paper is accepted, potentially using an additional page or an appendix to discuss the broader literature in speculative decoding and how their approach is orthogonal to some of the approaches, and for what methods it might not be readily applicable.
> >
> > I would also appreciate a more specific measurement or an in-depth discussion of the particular impact of the additional inference step compared to the vanilla cost in the paper, even if the focus of SD is on the inference cost. I believe it would be of interest to practitioners and for comparison with future works.

---

> ### Author Response · Authors · 2025-08-04
> **Thank you for the valuable review!**
>
> Thank you so much for the valuable review and for the Accept rating! We will include the related work discussion and additional cost analysis in the subsequent version.

---

### Official Review · Reviewer_ocoD · 2025-07-02

**Clarity:** 3
**Significance:** 3
**Originality:** 2
**Rating:** 4
**Confidence:** 3

**Summary:**

This paper proposes AdaSPEC, which demonstrates superior performance across diverse tasks (arithmetic reasoning, instruction-following, coding, summarization) and model configurations, consistently achieving higher acceptance rates than the state-of-the-art DistillSpec method Please refer to the strenths below.

**Questions:**

Please refer to the above weakness.

**Ethical Concerns:**

["NO or VERY MINOR ethics concerns only"]

**Final Justification:**

I have read the feedback from the authors. The issues seems to be resovled. Considering the responses from the authors and the comments from the other  reviewers, I would like to upgrade my rating to BA, the authors are encouraged to fully incorprate the concerns and improve the quality of the content in the paper during finalisation.

**Limitations:**

Yes.

**Quality:**

2

**Strengths And Weaknesses:**

## Strengths:

Consistent and Robust Experimental Performance: AdaSPEC demonstrates superior performance across diverse tasks (arithmetic reasoning, instruction-following, coding, summarization) and model configurations (31M/1.4B, 350M/2.7B), consistently achieving higher acceptance rates than the state-of-the-art DistillSpec method. This broad validation across different domains and model scales highlights the generalizability of the approach .

Innovative Selective Distillation Strategy: The introduction of selective token filtering via a reference model addresses a key limitation of existing knowledge distillation methods for draft models—their inability to fully align with target models due to capacity constraints. By focusing on "easy" tokens that the draft model can effectively learn, AdaSPEC improves token acceptance rates without sacrificing generation quality, representing a novel and intuitive solution .

Rigorous Ablation Studies: The paper conducts comprehensive ablation analyses on critical components, including token selection mechanisms, training methods, distillation objectives, and token selection ratios. These studies validate the effectiveness of each component (e.g., top 40% tokens outperform bottom 40%, forward KL is superior to TVD/RKL), enhancing the methodological credibility.

## Weaknesses:

Insufficient Support for the Reference Model: While the reference model is claimed to play a "crucial role" in filtering hard tokens, the paper lacks strong theoretical justification for why its perplexity-based filtering reliably identifies tokens beyond the draft model’s capacity. Additionally, there is a lack of direct experimental evidence (e.g., correlation between the reference model’s "hard token" labels and actual draft model errors) to validate its filtering accuracy .

Limited Comparisons to Recent Methods: The method is only benchmarked against DistillSpec (2023), without comparisons to more recent advancements in speculative decoding or knowledge distillation . This narrow comparison weakens claims of state-of-the-art performance and reduces the work’s timeliness .

Underdeveloped Generalization Analysis: While AdaSPEC shows promise in tested configurations, its generalization to larger model gaps (beyond 40x) or more complex, low-resource tasks is not thoroughly explored. The limitations section only briefly mentions potential improvements to filtering strategies, without addressing how the method might scale or adapt to diverse real-world scenarios .

---

> ### Author Rebuttal · Authors · 2025-07-30
>
> Dear Reviewer ocoD,
>
> Thank you for recognizing the potential and effectiveness of our work and for providing detailed constructive comments. Below, we address each point raised.
>
> ---
>
> >**W1**: Insufficient Support for the Reference Model: While the reference model is claimed to play a "crucial role" in filtering hard tokens, the paper lacks strong theoretical justification for why its perplexity-based filtering reliably identifies tokens beyond the draft model’s capacity. Additionally, there is a lack of direct experimental evidence (e.g., correlation between the reference model’s "hard token" labels and actual draft model errors) to validate its filtering accuracy .
>
> **R1**: Take GSM8K as an example, we argue that the reference model can actually prune out the hard tokens for the draft model:
>
> 1. An observation is that the hard tokens are typically math related tokens, as depicted in Appendix A.3, these tokens are difficult to predict as they involve mathematical logic, whose complexity is often beyond the draft model’s capacity.
> 2. Case Studies in Section 4.4 reveals that our token selection mechanism can successfully filter out hard tokens, especially those mathematical tokens.
>
> Besides, we need to point out that many researches on large models are driven by empirical results, with theoretical studies or justification lagging far behind empirical research. Our method has demonstrated strong results in practice and empirically supported our statement in how tokens are filtered, which we believe sufficiently substantiates our conclusions.
>
> ---
>
> >**W2**: Limited Comparisons to Recent Methods: The method is only benchmarked against DistillSpec (2023), without comparisons to more recent advancements in speculative decoding or knowledge distillation. This narrow comparison weakens claims of state-of-the-art performance and reduces the work’s timeliness .
>
> **R2**: As of now, DistillSpec is the only existing work that applies model distillation to train the draft model in the speculative decoding (SD) framework, and it is widely regarded as the state-of-the-art method in this line of research. Therefore, we believe our comparison to DistillSpec is both appropriate and meaningful within the scope of distillation-based SD methods.
>
> Moreover, our results demonstrate not only superior decoding efficiency under the standard SD architecture but also stronger generalization across models of different sizes. We have also highlighted that **our approach is orthogonal to other classes of SD techniques** (e.g.adaptive rejection, or sampling strategies), and can in principle be combined with them to further enhance performance.
>
> To further validate our proposed method on frontier SD approaches, we implemented AdaSPEC with Eagle—a representative SD method featuring tree attention that demonstrates superior performance to Medusa. Following the standard practice in the Eagle paper [1], we trained the Eagle decoding head on the ShareGPT dataset using the 3-epoch setting from our work and evaluated generation speed on MT-bench.
>
> *Table R1.* Vicuna v1.3 7B with 3-epoch ShareGPT. Here, train accuracy refers to first-generated-token accuracy in the training set.
>
> |                                 | **Eagle** | **Eagle + AdaSpec** |
> | ------------------------------- | --------- | ------------------- |
> | Train ACC ↑                     | 75.3%     | **76.3%**           |
> | Generation Speed (s/sentence) ↓ | 8.85      | **8.06** (-8.9%)    |
> | Generation Speed (tokens/s) ↑   | 63.48     | **68.21** (+7.45%） |
>
> These results demonstrate AdaSPEC's effectiveness and generalizability beyond vanilla SD to more advanced SD methods like Eagle, which enhances the practical utility and impact of our work.
>
> ---
>
> >**W3**: Underdeveloped Generalization Analysis: While AdaSPEC shows promise in tested configurations, its generalization to larger model gaps (beyond 40x) or more complex, low-resource tasks is not thoroughly explored. The limitations section only briefly mentions potential improvements to filtering strategies, without addressing how the method might scale or adapt to diverse real-world scenarios .
>
> **R3**: We would like to clarify that **we have already evaluated AdaSpec under a draft/target size ratio of 1/60** in our additional experiments on the **Qwen2.5-0.5B → Qwen2.5-32B** setting, which goes beyond the 1/40 ratio raised in the question. These results confirm that AdaSpec remains effective even under such extreme scale gaps.
>
> While we agree that analyzing even larger gaps could be interesting, we believe that in **general-purpose applications**, going beyond a 40× size ratio is rare and often **unrealistic**—particularly in settings where the draft model must still perform useful computations (e.g., general text generation). In practice, such extreme ratios may only arise in **domain-specific** or highly constrained deployment scenarios, which are somewhat outside the scope of our current work.
>
> In terms of **diversity**, we have already included a wide range of domains in our evaluation—spanning **mathematical reasoning, instruction following, code generation, text summarization**, and **multi-task mixtures** (as discussed in §4.6). In addition, our **dual setting design (3-epoch and optimal-epoch)** is intended to simulate **realistic training budgets and performance goals**, further supporting the robustness of AdaSpec across practical scenarios.
>
> [1] EAGLE: Speculative Sampling Requires Rethinking Feature Uncertainty

---

> > ### Comment · Reviewer_ocoD · 2025-08-07
> >
> > The responses on model gaps and domains seem to be reasonable. However, to better gauge practical impact, it is better to further showcase more distillation effects (e.g., detailed token - level distillation outcomes, comparative distillation efficiency across diverse complex tasks)? This would strengthen the case for real - world utility.

---

> ### Author Response · Authors · 2025-08-06
> **Kind Request for Further Discussions**
>
> Dear Reviewer ocoD,
>
> Thank you for your time and valuable feedback on our work. We hope that our response and revision have adequately addressed your concerns. As the discussion period nears its end, we would be very grateful if you could reconsider our work and possibly adjust the score accordingly. If you have any additional questions or suggestions, we would be happy to have further discussions.
>
> Best regards,
>
> Submission7301 Authors

---

> ### Author Response · Authors · 2025-08-07
> **Follow-up Response to Reviewer ocoD**
>
> Thank you for your follow-up. Below we address your requests regarding **token-level distillation outcomes** and **comparative distillation efficiency**, and clarify the practical implications of our method.
>
> ---
>
> #### **Token-Level Distillation Outcomes**
>
> Our paper **already provides a both qualitative and quantitative token-level analysis** to evaluate the impact of AdaSpec.
>
>
> In our GSM8K case study **(Appendix A.3 and Section 4.4)**, we show that in math domain, AdaSpec tends to select **math-related tokens** during filtering. These tokens represent structured, learnable reasoning patterns (e.g., numbers, mathematical operators, or logical steps). Our reference model helps focus the distillation process on these parts, enabling the draft model to better capture the reasoning logic through supervision from the target model.
>
> This aligns with our core idea: AdaSpec guides the draft model to allocate capacity where it matters most—**on tokens that are important and learnable** but may otherwise be under-emphasized in uniform distillation. This observation offers **token-level evidence**.
>
> In addition to case-level analysis, we also provide **statistical token-level evidence** across datasets in Figure 2, which supports the effectiveness of AdaSpec’s selective distillation approach:
>
> - **Logit Margin Distributions:** AdaSpec increases both the frequency and magnitude of positive top-2 logit margins while reducing negative ones, showing the draft model makes **more confident and accurate** predictions.
>
> - **KL Divergence Across Tokens:** AdaSpec yields consistently lower token-level KL divergence, with a distribution shift toward zero compared to DistillSpec, indicating **tighter alignment** with the target model’s outputs.
>
> Together, these metrics offer a multi-faceted view of token-level alignment, confidence, and knowledge transfer quality.
>
>
> ---
>
> #### **Comparative Distillation Efficiency across Diverse Tasks**
> In terms of task diversity, we **have already** included a wide range of domains in our evaluation—spanning mathematical reasoning, instruction following, code generation, text summarization, and multi-task mixtures (as discussed in **§4.6**).
>
> Moreover, our dual setting design—comparing 3-epoch and optimal-epoch training—serves to simulate varying training budgets and practical constraints, providing further evidence of AdaSpec’s robustness across different real-world use cases.
>
> Importantly, the **token-level analyses** (e.g., logit margin distributions and KL divergence metrics) and the **ablations in §4.5** are also conducted across these diverse tasks. This reinforces that the gains from AdaSpec are not task-specific artifacts, but hold across multiple domains and model behaviors.
>
> ---
>
> #### **Practical Impact and Scalability**
>
> We believe AdaSpec offers real-world utility based on the following:
>
> 1. **Easy integration**: It is implementation-friendly and requires no architecture changes.
> 2. **Scalable**: We validated it with up to a **60× model size gap** (e.g., Qwen2.5-0.5B → Qwen2.5-32B), showing strong performance even in extreme settings. While >70B experiments are out of academic reach, our method poses **no intrinsic barrier to industrial-scale deployment**.
> 3. **Orthogonal to other methods**: As shown with **Eagle + AdaSpec**, our method can be combined with orthogonal improvements (e.g., adaptive sampling, tree attention) for **additional speedup (~10%)**.
> 4. **Superior to DistillSpec**: Compared to DistillSpec—the only other distillation-based speculative decoding baseline—**AdaSpec achieves 10–20% faster inference** under the same standard speculative decoding framework.

---

### Official Review · Reviewer_fMLq · 2025-07-02

**Clarity:** 3
**Significance:** 3
**Originality:** 3
**Rating:** 5
**Confidence:** 4

**Summary:**

This submission introduces AdaSPEC, a selective knowledge distillation approach for optimizing efficient LLM inference via speculative decoding. AdaSPEC, initially relies on DistillSpec framework to distill a reference draft model from the original (verification) model. Subsequently, "easy" tokens are identified based on their loss, and selected to train the draft model of the resulting SD pipeline. The intuition behind the proposed approach is that by excluding the difficult tokens from the distillation process for fitting the small-capacity draft model, its training process will become less ambiguous and converge to a more robust and effective draft model (i.e. lead to higher acceptance rate and thus speed-up for the overall SD pipeline). The provided experimental results validate this assumption, showcasing consistent gains across different model pairs and datasets.

**Questions:**

Please considering replying on the points raised in the comments section above.

**Ethical Concerns:**

["NO or VERY MINOR ethics concerns only"]

**Final Justification:**

Increasing my score from 3 to 4, as the raised concerns have been satisfactorily addressed by the authors during the rebuttal phase.
The proposed methodology is demonstrated to have notable contribution, novelty and effectiveness and its publication can benefit the community, in my opinion.

**Limitations:**

Some limitations of the proposed approach are discussed at the conclusions section, along with proposed future work directions.

**Paper Formatting Concerns:**

No major formatting concerns.

**Quality:**

3

**Strengths And Weaknesses:**

Strengths:
- The proposed approach is very interesting and timely and exploits an interesting insight.
- The manuscript is well written and easy to follow.
- The effectiveness of the proposed methodology is demonstrated through extensive experiments across varying tasks and model pairs.
- The conducted ablations offer potentially valuable insights for applying KD in related SD settings.

Comments:
- It is unclear why the proposed filtering approach (eq 9) evaluates and compares the loss of the reference vs the draft model (Mq), instead of the correctness/distance of the reference (distilled) model to the original (Mp). Since Mq is the model being trained in step2, Mq's perception of easy samples is expected to change overtime, however the proposed approach does not appear to adapt to such changes. In contrast, evaluating the correctness of Mref (w.r.t Mp) may potentially form a more robust and stable proxy of the predictive difficulty of each token for the draft model.
- It is unclear how the effectiveness of the proposed approach is affected by the learning capacity of the draft model (and whether the percentage of selected tokens should be scaled accordingly). Although the authors have conducted experiments between 2 different pairs of models, with varying relative parameter counts, more robust insights about the scalability of the proposed method will largely benefit the community and enhance the applicability of the proposed approach.

---

> ### Author Rebuttal · Authors · 2025-07-30
>
> Dear Reviewer fMLq,
>
> Thank you for recognizing the potential and effectiveness of our work and for providing detailed constructive comments. Below, we address each point raised.
>
> ---
>
> >**W1**: It is unclear why the proposed filtering approach (eq 9) evaluates and compares the loss of the reference vs the draft model (Mq), instead of the correctness/distance of the reference (distilled) model to the original (Mp). Since Mq is the model being trained in step2, Mq's perception of easy samples is expected to change overtime, however the proposed approach does not appear to adapt to such changes. In contrast, evaluating the correctness of Mref (w.r.t Mp) may potentially form a more robust and stable proxy of the predictive difficulty of each token for the draft model.
>
> **Reply**:
>
> As we propose in "introduction" chapter (line 57), our goal here is to identify tokens that are **more learnable** at the current training stage of the draft model Mq. Compared to the absolute value of per-token loss/accuracy, **the relative loss change is more indicative of ease of “learnability”**. This is because the potential to learn each token varies, and **a high loss/low accuracy does not necessarily equate to ease of learning**. Evaluating the difference between Mref and Mp, as suggested, would instead reflect how closely the reference approximates the teacher model, but may not align with the objective of selecting *learnable* tokens for improving Mq, leading to suboptimal results.
>
> We think that possibly the reviewer might be referring to the initial (untrained) version of Mq when mentioning Mp. Interpreted this way, comparing to the base model could provide an alternative view on token difficulty, and we agree that this is an interesting direction worth exploring in future work. That said, this may be more likely to be viewed as an alternative design choice rather than a clear weakness.
>
> ---
>
> >**W2**: It is unclear how the effectiveness of the proposed approach is affected by the learning capacity of the draft model (and whether the percentage of selected tokens should be scaled accordingly). Although the authors have conducted experiments between 2 different pairs of models, with varying relative parameter counts, more robust insights about the scalability of the proposed method will largely benefit the community and enhance the applicability of the proposed approach.
>
> **Reply**:
>
> **The first question is the capacity of the draft model.** Technically, the capacity of the draft model is relative to the target model, which can be depicted by the size gap between them. The two pairs of draft/target size ratios are representative in contemporary research : $\sim$1/40 for Pythia setting, and $\sim$1/8 for Phi-2 setting. This holds practical significance in real-world applications.
>
> When practical applications require larger target and draft models, our method is designed to be implementation-friendly and readily scalable. It can be integrated seamlessly into existing training pipelines without architectural changes, making it suitable for scaling to larger models and datasets.
>
> In **Section 4.6**, we conducted further GSM8K tests using a combination of Qwen2.5-0.5B and Qwen2.5-32B models ($\sim$1/64 size ratio), which demonstrate that our approach remains effective when scaling up to larger models. We again present those results below.
>
> *Table R1.* Acceptance Rate of Qwen2.5-0.5B$\rightarrow$32B with 3-epoch GSM8K.
>
> |       | **DistillSpec** | **AdaSpec** |
> | ----- | --------------- | ----------- |
> | GSM8K | 84.43%          | 86.21%      |
>
> **The second question is how to scale k.** This hyperparameter appears to be **relatively stable with respect to model scale**, and we did not find strong evidence that it needs to be adjusted as the model size increases. Thus, we keep it to 0.4 for all shapes of the models, even for the Qwen2.5-32B experiment.
>
> While we acknowledge that experiments on even larger models (e.g., >70B) would be valuable, such experiments are unfortunately beyond the computational affordability of academic research. Nonetheless, our method poses no intrinsic barrier to such scaling, and we believe it can be readily adopted in industrial-scale settings.
>
> **The third question is the generalizability of AdaSpec on broader settings and applications.** To further validate AdaSpec on more advanced approaches beyond vanilla SD, we have now included additional experiments combining our method with Eagle, a popular decoding strategy utilizing tree attention for multi-token speculation. Following the standard practice in the Eagle paper [1], we trained the Eagle decoding head on the ShareGPT dataset using the 3-epoch setting from our work and evaluated generation speed on MT-bench.
>
> *Table R2.* Vicuna v1.3 7B with 3-epoch ShareGPT. Here, train accuracy refers to first-generated-token accuracy in the training set.
>
> |                                 | **Eagle** | **Eagle + AdaSpec** |
> | ------------------------------- | --------- | ------------------- |
> | Train ACC ↑                     | 75.3%     | **76.3%**           |
> | Generation Speed (s/sentence) ↓ | 8.85      | **8.06** (-8.9%)    |
> | Generation Speed (tokens/s) ↑   | 63.48     | **68.21** (+7.45%） |
>
> The results in the table above demonstrate AdaSPEC's effectiveness and generalizability beyond vanilla SD to more advanced SD methods like Eagle.
>
> [1] EAGLE: Speculative Sampling Requires Rethinking Feature Uncertainty

---

> > ### Comment · Reviewer_fMLq · 2025-08-04
> >
> > I appreciate the authors' thorough responses to all raised comments, which satisfactorily address my concerns.
> > Having read the rebuttal as well as the other reviewers' comments and relevant discussion, I believe this submission proposes an approach with notable novelty and contribution to the current SoTA. As such, I am inclined to increase my score to Accept.

---

> > > ### Author Response · Authors · 2025-08-05
> > > **Thank you for raising the score!**
> > >
> > > Thank you so much for the Accept rating! We will revise our article based on your valuable feedback.

---

### Note · Authors · 2025-08-12

We sincerely thank all reviewers for their insightful, constructive, and thoughtful feedback. Below we briefly summarize the key points of the discussion.

**Key design choices in the paper.** We are pleased that reviewers praised the novelty of our approach. Besides, we believe that all the concerns and questions are addressed, including:

1. The rationale behind our filtering mechanism (based on relative learning progress rather than absolute correctness).
2. The stability of the filter ratio k across model scales, and the distinction between our method and prior works like Rho-1.

**Practicality, generalizability, and scalability of AdaSPEC.** While the consistency of results has been recognized by the reviewers, some reviewers are concerned about AdaSpec's real-world utility. We address this with:

1. We have validated AdaSPEC under both constrained (3-epoch) and optimal-budget training, confirming its robustness across diverse real-world use cases.
2. Our main experiments are comprehensive, covering a wide range of domains including mathematical reasoning, instruction following, code generation, text summarization, and multi-task mixtures.
3. We further test AdaSPEC with frontier $\sim$30B model and under extreme model size ratios (up to 1/64), achieving 84.43%→86.21% improvement for acceptance rate, demonstrating its stability even in large-scale capacity gaps.
4. We conduct comprehensive token-level analyses (**logit‑margin right‑shift** and **KL left‑shift**, and case studies where AdaSpec’s errors are nearly a subset of baseline errors) as well as ablations to substantiate the quality and generality of the distillation.
5. Actual wall clock speedup is provided, demonstrating that **AdaSPEC achieves measurable performance gains (up to 20%)** compared with DistillSpec in real-world deployment scenarios.
6. Within **distillation‑based SD**, DistillSpec is the only directly comparable SoTA. Furthermore, integrated with advanced SD methods such as **Eagle**, AdaSpec achieves up to **8.9% faster generation speed** and **7.45% higher token throughput**.

We emphasize that AdaSPEC is a lightweight, plug-and-play enhancement to existing SD pipelines—with no architectural changes and only marginal training overhead. The **persistent inference speedup** outweighs this one‑time cost in efficiency‑critical deployments, which **directly advances the core goal of SD**. This makes it a highly practical solution for deployment in real world settings.

---

### Decision · Program_Chairs · 2025-09-17

**Decision:**

Accept (spotlight)

**Comment:**

This paper introduces AdaSPEC, a novel selective knowledge distillation method for optimizing speculative decoding in large language models. The key contribution is a two-phase distillation process that uses a reference model to identify and filter out "hard-to-learn" tokens, allowing the draft model to focus its limited capacity on mastering "easier" tokens. This approach leads to better alignment between the draft and target models, resulting in higher token acceptance rates and, consequently, faster inference speeds. The method is evaluated across a range of tasks, including arithmetic reasoning, instruction-following, coding, and summarization, demonstrating consistent improvements over the state-of-the-art DistillSpec method.

The paper received consistently positive reviews, with final ratings of 5, 5, 5, 4, 4. Reviewers generally expressed high confidence in their assessments, with scores. There were no significant conflicting viewpoints among reviewers; initial concerns were addressed during rebuttal, and several reviewers raised or confirmed positive scores. All agreed on the paper's strengths and the importance of its contribution.

All reviewers pointed out the novelty and timeliness of the proposed approach, the clarity of the writing, and the extensive experimental validation across various tasks and model configurations.

Still, there were some initial concerns included the modest empirical results in some settings, the additional computational cost of the two-phase distillation process, and the need for a more in-depth comparison with other speculative decoding methods.

While not a theoretical paper, the underlying intuition of focusing on "learnable" tokens is well-motivated and supported by the empirical results. The experimental evaluation is thorough and well-executed, with a diverse set of tasks and model pairs. The ablation studies provide valuable insights into the effectiveness of each component of AdaSPEC. The paper also provides sufficient detail for reproducibility, including hyperparameter settings and a clear description of the methodology.

This work has the potential to significantly impact the field of efficient LLM inference by providing a practical and effective method for improving speculative decoding. Furthermore, AdaSPEC is a plug-and-play enhancement to existing speculative decoding pipelines, making it highly practical for real-world deployment.

This is a technically solid paper with a novel and impactful contribution to the field of efficient LLM inference. The proposed method, AdaSPEC, is well-motivated, thoroughly evaluated, and demonstrates consistent improvements over the state-of-the-art. The authors have done an good job of addressing the reviewers' concerns and have made a strong case for the practical applicability of their work.

This paper will be of great interest to the NeurIPS community and I am pleased to recommend it for a spotlight presentation.